# Battery-Less Environment Sensor Using Thermoelectric Energy Harvesting from Soil-Ambient Air Temperature Differences

**DOI:** 10.3390/s22134737

**Published:** 2022-06-23

**Authors:** Priyesh Pappinisseri Puluckul, Maarten Weyn

**Affiliations:** IDLab—Faculty of Applied Engineering, University of Antwerp—imec, Sint-Pietersvliet 7, 2000 Antwerp, Belgium; maarten.weyn@uantwerpen.be

**Keywords:** battery-less, DASH7, energy harvesting, soil temperature, thermoelectric effect, thermoelectric generator

## Abstract

Energy harvesting is an effective technique for prolonging the lifetime of Internet of Things devices and Wireless Sensor Networks. In applications such as environmental sensing, which demands a deploy-and-forget architecture, energy harvesting is an unavoidable technology. Thermal energy is one of the most widely used sources for energy harvesting. A thermal energy harvester can convert a thermal gradient into electrical energy. Thus, the temperature difference between the soil and air could act as a vital source of energy for an environmental sensing device. In this paper, we present a proof-of-concept design of an environmental sensing node that harvests energy from soil temperature and uses the DASH7 communication protocol for connectivity. We evaluate the soil temperature and air temperature based on the data collected from two locations: one in Belgium and the other in Iceland. Using these datasets, we calculate the amount of energy that is producible from both of these sites. We further design power management and monitoring circuit and use a supercapacitor as the energy storage element, hence making it battery-less. Finally, we deploy the proof-of-concept prototype in the field and evaluate its performance. We demonstrate that the system can harvest, on average, 178.74 mJ and is enough to perform at least 5 DASH7 transmissions and 100 sensing tasks per day.

## 1. Introduction

Internet of Things (IoT) and Wireless Sensors Networks (WSN) are often value for the elegance and easiness they bring to remote parameter monitoring and sensing applications. The ability to operate without any human supervision and assistance is one of the key features that makes them the first choice for many remote sensing and measurement applications. In applications where frequent access to the installation site is costly and difficult, these systems provide an attractive solution. Despite being famous for their ultra-low-power footprint, IoT and WSN devices are still limited in terms of their lifetime in many application scenarios due to the nature of the energy sources used. In most cases, the energy sources are a primary or secondary battery. Primary batteries are limited to one-time uses, whereas secondary batteries need a nearby energy source for recharging. Energy harvesting has proven to be an important tool in prolonging the lifetime of IoT and WSN systems, as it can provide an energy top-up by scavenging energy from ambient sources such as solar, wind, heat, etc.

Solar energy is a commonly used energy harvesting source for outdoor wireless sensors. It has the advantage of high efficiency and availability. However, there are situations where the choice of solar harvesters is not practical. For instance, in environmental sensing, devices are deployed in the middle of a dense vegetation or even close to the soil surface. In such cases, solar cells are prone to decayed efficiency due to the shadow of vegetation and the dust cover that accumulates over time. On the other hand, thermoelectric generators (TEGs) depend only on the temperature difference between the cold and hot junctions. The Seebeck effect explains the phenomenon behind TEG devices. The use of TEGs for energy harvesting is not a new idea. There has already been a lot of research showcasing the effective use of the TEG in enabling the long-term operation of IoT devices [1,2,3,4]. A sustained operation of the TEG harvesters mostly depends on the temperature difference across its junctions. Usually, a heat sink is attached to one side of the TEG, so that maximum heat transfer happens. Since the operation of TEG devices depends only on the temperature difference, energy can be harvested from any environment where a temperature gradient is present, for instance, from a water-air interface that is present in aquatic environments [5]. Similarly, a soil-air interface can also be a potential choice for thermal energy harvesting. It is a known fact that the temperature of the soil varies rather slowly compared to the daily ambient temperature and, at depths of more than 30 cm, the temperature is approximately constant [6]. Hence, there is always a temperature gradient between the soil and nearby air. For instance, we have been recording the soil temperature data from the Campus Drie Eiken (CDE) of University of Antwerp and the Forhot [7] research site in Iceland, as presented in Figure 1a,b respectively. Temperature data from both the sites present a clear delta between soil and air temperature. This difference in temperature can be utilized to produce energy to power low-power wireless sensor nodes.

The number of connected sensing devices installed around the world is increasing, as is the number of batteries. It is estimated that, by 2025, almost 30.9 billion wireless connected devices will be installed [8]. This contributes to approximately 78 million batteries being thrown into the bin every day [9]. Primary cells end up in the bin immediately after their first use. In many cases, they are pro-actively replaced to ensure a continuous operation, whereas the secondary batteries become useless after a certain number of charge-discharge cycles. The average lifetime of the batteries ranges from 3–10 years after which they end up as hazardous waste in the dump. Moreover, these batteries are often not recycled. According to Perchards, 56% of discarded batteries in Europe are not properly recycled [10]. Thus, there is a demand for systems that are sustainable and make no or less contribution to e-waste. This is one of the major driving forces for battery-less sensing systems. In a battery-less system, batteries are replaced with ultra- or super-capacitors, which offer many advantages over batteries. One of their main advantages is that their lifetime is hardly affected by the charge-discharge cycles. Additionally, their characteristics are least altered by operating temperature. Therefore, they can be used in a wide range of operating temperatures.

In this paper, we present a proof-of-concept, battery-less sensing system for environmental sensing. The device harvests energy from soil-air temperature differences with the help of a TEG device. The battery-less system can measure soil temperature, ambient temperature and humidity and use DASH7 [11] to achieve wireless connectivity. Our research in this domain is mainly motivated by the environmental sensing and monitoring that is to be performed at the Forhot [7] site in Iceland. The Forhot site presents naturally warm soil due to geothermal activity. The temperature of the soil at different locations in Forhot observed to date ranges from 0 °C to 50 °C [12].

We make the following contributions in this paper:1We present soil temperature, air temperature and TEG open-circuit voltage collected from fields in Belgium and Iceland. We use these collected data to simulate energy output from a TEG harvester.2We design an end-to-end, battery-less, wireless sensor node for environmental monitoring. The design and integration of energy harvesting, power management and storage circuit, sensor interface and wireless communication module are discussed.3We integrate, for the first time, DASH7 wireless communication with soil thermal energy harvesting, and enable battery-less operation.4We provide a real-world evaluation of the system and the data collected from field experiments.

The remainder of the paper is organized as follows. Section 2, presents TEG’s background and the state-of-the-art of soil thermal energy harvesting, along with a brief introduction to the DASH communication protocol. Section 3, presents the soil and air temperature data that were collected from the sites in Iceland and Belgium. An estimation of harvester energy based on the collected data is discussed. In Section 4, the sensor prototype design is described. Section 5, presents a discussion and analysis of the data collected from the field installation of the prototype. Finally, we discuss the future work in Section 6, and conclusions are drawn in Section 7.

## 2. Related Work

In this section, we provide the reader with the relevant state-of-the-art on TEG and DASH7.

### 2.1. A Brief on Thermoelectric Generators

Thermo Electric Generators (TEG) are solid-state devices that can convert heat energy into electricity. The Seebeck effect explains the principle of operation of TEG. The Seebeck effect says that a carrier movement in dissimilar materials can be triggered by a temperature difference. This leads to the conversion of heat energy into electrical energy. A TEG usually consists of an array of P- and N-type semiconductor materials. The materials are thermally connected in parallel and electrically connected in series. A symbolic image of a TEG module is shown in Figure 2. When a temperature difference is applied across the junction, a potential difference is produced. The holes in the P junction diffuse to the N side and the electrons in the N region diffuse to the P side. A TEG is a combination of multiple such P-N elements. The two sides of the TEGs are usually referred to as the cold side and the hot side.

The output voltage produced by a TEG depends on the delta temperature between its hot and cold sides. Therefore, one of the major design requirements when using a TEG is the efficient transfer of heat from the hot side to the cold side, so that the maximum temperature difference is present across the junctions. Usually, a heat sink is attached to one of the TEG sides to dissipate heat. The other side is attached to the heat source.

Powering IoT devices and wireless sensor nodes through TEG energy harvesting is a widely researched area. The most common application is in industrial environments where waste heat from machines or furnaces is converted into energy [13]. These industrial use cases usually can provide a high-temperature delta across TEG junctions. It is often challenging to use TEGs in low-temperature-difference situations due to their reduced efficiency. However, researchers have already proven that TEGs, although with compromised efficiency, can still generate enough energy to power wireless nodes or devices. One of the instances of a low-grade thermal energy harvesting system is presented in [14], where the authors use the temperature gradient in asphalt to generate energy. Following this, there have been many works that exploited the possibility of harvesting energy from the asphalt and road temperatures [15,16,17].

Similarly, some researchers have demonstrated the possibility of harvesting energy from ambient-soil temperature differences. Ref. [18] presents one of the earlier attempts in this area. They used a system with a buried heat sink to convert heat energy to electrical. Based on the temperature measurements taken from the fields, they calculate a peak power of 0.4
mW from a 60-h experiment. In another work [19], an energy-harvesting system with multiple TEG modules that can convert diurnal heat flow through the upper-soil layer into energy is presented. The authors claim that they were able to produce a peak power of 9.2
mW with four thermocouples connected in series.

In [20], researchers evaluated soil energy harvesting based on the temperature difference data collected at different soil depths. They further simulated a target node model with thermal energy harvesting and showed that the target node can transmit packets from 7.5–40 packages per h.

The ground temperature harvesting system presented in [21] can generate a maximum power of 27.2 mW in the night and 6.3 mW during the daytime. The authors use a novel auto-polarity detection circuit to deal with the polarity reversal of TEG voltage in night time. They also provide data from a multi-year-long evaluation of their system in the field.

A study on the harvesting of energy from forest soil is presented in [22]. They claim a harvested output of 128.74 J in a single day and 5209.92 J for the duration of the experiment.

### 2.2. DASH7—A Medium-Range Communication Protocol

DASH7 is a medium-range communication protocol that operates in the 868 MHz ISM band. DASH7 is based on the ISO/IEC 18000-7 standard for RFID tags and sensor nodes [11]. DASH7 supports a star network topology and is intended for the short and quick transfer of data from an end-node to a gateway [23]. Since the protocol is based on the RFID standards, most of the communication over DASH7 is modelled as file operations. DASH7 supports two modes of communication: the PUSH and PULL mode. The PUSH mode is used when the endpoint spends most of its time in sleep and transmits only when data are available. The endpoint can request an acknowledgement for the transmitted from the gateway. One of the key features of DASH7 is its support of a downlink data transfer with the so-called PULL mode. In PULL mode, the endpoint wakes up at predefined intervals and sniffs the channel to see if there is any incoming file command from the gateway. An ad-hoc synchronization mechanism allows the endpoint to synchronize with the gateway and receive wake-up calls.

DASH7 physical layer employs the Gaussian Frequency Shift Keying (GFSK) modulation scheme to modulate the data. The MAC layer can perform a polite spectrum access with the help of Clear Channel Assessment (CCA). A DASH7 device can be configured to use either a low-rate (LO_RATE), normal-rate (NO_RATE) or high-rate (HI_RATE) data rate. We have shown that the power consumption of DASH7 is comparable with the LoRAWAN module [24]. We used the same wireless module as in [24], which can be configured to work either as a DASH7 modem or as LoRaWAN device.

## 3. Field Evaluation—Soil Temperature and Air Temperature

Previous studies prove that the temperature of soil varies rather slowly with the daily air temperature [6]. In fact, the temperature at depths higher than 10 cm is least impacted by the daily air-temperature variations. To evaluate the possibility of generating power from soil temperature to drive a wireless sensor node, we conducted real-world experiments. We deployed a testbed to continuously monitor the soil temperature, air temperature and TEG open-circuit voltage. The testbeds were deployed in two locations: one in the Campus Drie Eiken (CDE) of the University of Antwerp (51.161° N, 4.408° W) in Belgium and the other in the Forhot research site (64.008° N, 21.178° W) in Iceland. The Forhot site offers a special case of soil temperature, as the soil is geothermally heated. The soil temperature varies from plot to plot with the currently measured minimum being 0 °C and the maximum measurement being 50 °C [12].

The temperature of the soil, air and TEG open-circuit voltage were recorded. Temperatures were measured with 1-wire DS18B20 [25] temperature sensors and the open-circuit voltage was measured with a 12 bit ADC. TG12-4-01LS thermoelectric generator [26] from Marlow industries was used as the generator. A copper rod of 15 cm was used to provide a heat-transfer path between the soil and the cold side of the TEG. A heat sink ATS55300K-C1-R0 [27] was connected to the hot side of the TEG. Measurements were taken every five minutes and transferred to a cloud storage for real-time view and analysis.

### 3.1. Data Collected from CDE Campus, University of Antwerp

Data collected from the CDE campus for a duration of 47 days are presented in Figure 3. From the measurements at CDE, we can observe that the soil temperature varies relatively slowly with the air temperature, but an average daily fluctuation of ±2 °C is observed in soil temperature at 15 cm depth. The TEG generates voltage almost all the time, with the highest output being produced in the daytime. Further, we observe a blackout period in TEG output with zero- or minimum-output voltage. On average, this lasts for around 6 h per day, and happens during the dawn and dusk, when the air temperature is equal to the soil temperature. Additionally, the output voltage produced at night is comparatively smaller than that produced during the daytime.

### 3.2. Data Collected from Geothermally Active Forhot Site in Iceland

The data collected from the Forhot site present a more interesting dataset due to the fact that the soil there is geothermally active. Additionally, the variation in soil temperature with diurnal variation is very low; the soil at different plots is warmed to a different temperature. At present, there are plots identified with a temperature of 0–50 °C [12]. During the experiments, the hot side of the TEG was attached to the copper rod and the cold side was connected to the heat sink. We collected data from a plot, which was warmed to 20 °C at 10 cm depth. Temperature measurements and TEG open-circuit voltage are presented in Figure 4.

We observe from the measurements that the variation in soil temperature with the ambient temperature is relatively low or negligible. Moreover, the fluctuations in the open circuit voltage are less compared to the data from CDE. Since the soil temperature is always higher than the air temperature, a polarity reversal is also not present in the output voltage. However, we observed that in contrast to the data from CDE experiments, the temperature of the cold side of the heat sink in Forhot differs significantly from the air temperature. This can be attributed to the efficiency of the heatsink-TEG interface. Further, the device was mostly under snow during the whole measurement period. Despite these differences, the TEG is still able to produce a relatively significant output during the whole experiment duration.

### 3.3. Estimation of Energy

From the data collected from CDE and Forhot sites, we estimate the amount of energy that can be harvested by the TEG. For this, we use the open circuit voltage collected from the fields. We excite the TEG to produce the same open-circuit voltage values as those collected from the field. Then, the TEG output is converted to 4.2 V using a power management board [28]. This voltage is used to charge a capacitor and the charging current and the voltage levels of the capacitor are noted. From these values, the energy output is calculated for both the Forhot and CDE sites. The estimated energy values are presented in Figure 5.

Figure 5a shows the harvested energy calculated from the CDE data. The simulation predicts a maximum of 1.8 J. However, blackout days where the energy level is considerably low are present. For instance, on 29 April, the energy produced is almost zero. This is also visible in the open-circuit voltage data. The average energy produced for the experiment duration is 498.91 mJ, whereas the data produced from the Forhot site, shown in Figure 5b, have a near-consistent energy production, with an average of 2.017 J, over the whole experiment. Based on these energy values, we can observe that the thermal harvesting system, both in CDE and Forhot, can power a DASH7 node without trouble. The DASH7 node consumes only 11.5 mJ for the transmission of a packet [24].

## 4. Sensor Design

The design process of a temperature-monitoring sensor powered by the soil-temperature difference involves developing an energy converter unit, a power- management and -monitoring circuit and a sensing and radio unit. The energy converter unit employs a TEG module to extract the temperature difference between the soil and the air into electrical energy. The converted energy is transformed into a range that is suitable for charging an energy storage unit. The power management circuit keeps track of the incoming energy and alerts the sensing unit when there is enough energy for its operation. The energy from the storage unit is used to power the sensing and radio unit, which includes a DASH7 transmitter, ambient temperature and humidity sensor and a soil-temperature sensor.

### 4.1. Energy Converter Unit

The energy converter unit is designed to harvest the maximum amount of energy by providing an efficient heat transfer between the hot and cold sides of the TEG. We follow a traditional design, where a TEG is sandwiched between a copper rod and a heat sink. The hot side makes contact with the ambient air through a heatsink. We used the heat sink ATS-55300K-C1-R0 from Advanced Thermal Solutions Inc and the TEG module TG12-4-01LS from Marlow Industries. The TEG has a dimension of 300 × 340 mm. To ensure minimum thermal resistance, the interface between the heat sink and TEG module was sealed with thermal paste. The cold side of the TEG was connected to a copper road of 6 mm in diameter and 15 cm in length. The copper rod acts as a heat sink or source depending on the hot-side temperature. Copper was chosen as the heat tube due to its high thermal conductivity. The cold-side-copper interface was connected with thermal paste. The copper rod was further covered with a PVC pipe filled with a thermal insulator. Only 2 cm portion of the copper rod was exposed to the outside. This was performed to ensure that the copper rod does not come into contact with the upper soil layers, where the temperature fluctuations are high. The TEG module was then enclosed in a hollow enclosure made with acrylic. The heatsink and copper rod extended to the outside environment, as shown in Figure 6. The internal cavity of the acrylic enclosure was filled with thermal insulators to ensure minimal heat flow through the walls.

There are different points that we considered while designing the energy converter unit’s architecture. One of the major points was the energy-efficiency and reputability of the energy converter unit. Since the harvested energy mostly depends on the efficiency of the energy-converter unit, it is essential that we maximize its output. We can easily increase the harvested energy by connecting multiple TEG modules in series or parallel. Additionally, we can increase the length of the copper rod so that it penetrates deeper, allowing us to obtain a higher delta temperature. However, this has a direct impact on the Bill of Materials (BoM). The addition of an extra TEG module would increase the device cost by approximately 21 €, whereas increasing the length of the heat pump to reach deeper soil will add an extra 10 € to the cost, when considering copper as the heat pump. Further, as we plan to deploy more of these devices in future, we must obtain a simple yet efficient architecture that is cost-effective and can easily produce multiple units. A higher number of TEGs and longer copper rods will directly impact the form factor. The efficiency of the energy converter unit can be further improved by using a Phase Change Material (PCM) [1], which helps to maintain a constant temperature at the TEG sides. However, we do not rely on this method, for two reasons: one, when using a PCM, it is essential that we choose one that is suitable for the application’s temperature range. However, in our case, the temperature range can vary, largely due to the seasonal changes as well as the relocation changes. Hence, it is difficult to choose the right PCM. Secondly, the energy converter unit requires a complex design to ensure that the PCM does not come into direct contact with the environment. For instance, a PCM such as water, if used, would easily evaporate over time. Finally, we want to ensure that the overall system design uses an easily available material so that the fabrication of multiple units is possible without much trouble.

### 4.2. Power Management Unit

Th power management circuit plays a vital role in any energy-harvesting system. Since the temperature difference across the TEG plates is normally low, the output power generation is limited to micro-watts most of the time. Moreover, the output voltage falls in the mV range. Thus, we must employ a highly efficient power management system, which can boost the input voltage in mV into the voltage range with minimal loss, so that a storage element can be charged. We designed a simple yet efficient power management system, which uses minimal components to reduce the quiescent current consumption to the lowest possible value. We designed a power management system based on Matrix Industries’ boost converter, Mercury MCRY12 [29]. Mercury converters are designed to work with extremely low-input voltage sources. The converter’s *transformer reuse* technology enables it to work with input voltage levels of as low as 18 mV and a cold start voltage of 24 mV. A block diagram of the power management unit is shown in Figure 7.

We used a 15 mF super-capacitor [30] to store the converted energy (C_STORE_). The supercapacitor was charged with the 4.2 V boosted output from the Mercury converter. An ultra-low, quiescent, current, buck-boost converter TPS63900 from Texas Instruments [31] was then used to regulate the supercapacitor voltage to 3.3
V. From the datasheet, the quiescent current consumption of the regulator is 75 nA, and this was verified with a Joulescope [32]. A buck-boost regulator was necessary, since the voltage of the supercapacitor is not constant. The boost section ensured that the output was regulated for all the input voltages that are below the output level, whereas the buck ensured that any input levels above the output level were regulated. TPS63900 can regulate input voltage down to 1.8
V and provide 3.3
V for the load. The switching regulator also has an enable pin, which can be used to turn it on or off. A nano-watt comparator with hysteresis, designed using TLV3691 [33], monitors the input voltage and drives the enable pin. This ensures that the regulator is only activated when there is enough energy in the supercapacitor. At present, the lower threshold of the comparator is fixed at 2.4
V and the higher threshold at 3.3
V. These thresholds were chosen based on the energy consumption of the sensing and radio unit for one cycle of sensing and data transmission.

In addition to the main storage capacitor, we used a 500 μF capacitor as a bootstrap capacitor (C_BOOT_). The voltage from this capacitor was used to power the comparators and provide them with stable reference voltage. The bootstrap capacitor further controls the charging of C_STORE_ through the switch S2. S2 is an ultra-low, quiescent current load switch SiP32431 [34] from Vishay Semiconductors. SiP32431 has a quiescent current of 10 pA and leakage current of 200 pA. The active high enable input of SiP32431 is controlled by V_BOOT_. This bootstrap system ensures that the comparator circuits obtain a stable reference voltage independent of C_STORE_.

### 4.3. Sensing and Radio Unit

The sensing and radio unit comprises an ambient temperature sensor, a 1-Wire soil temperature sensor and a DASH7 transceiver module (Figure 8). We used SHT31A-DSI-B [35] to sense the ambient temperature and humidity and DS18B20 [25] to sense the soil temperature. We used the same wireless module as in [24], which is based on Murata ABZ [36] and runs the Sub-IoT stack for DASH7 [37]. The sensors are interfaced with the DASH7 module and this can trigger the sensor measurements and perform data transmission. The whole sensing and transmission unit can be powered with 3.3
V. The unit consumes 900 μJ energy for sensing and 11.5
mJ energy to transmit 6 bytes.

All three modules were further connected and enclosed inside an IP66 ABS enclosure for deployment, as shown in Figure 9.

## 5. Results

The energy-harvesting unit, along with additional data loggers to monitor system performance, was installed in the CDE campus for real-life evaluation. A block diagram of the whole experiment setup is depicted in Figure 10. We collected data from the energy harvesting device for around a month starting from 1 March 2022 to 12 April 2022. The data loggers measured supercapacitor (C_STORE_) voltage every one minute, along with the temperatures, and forwarded this information to a DASH7 gateway installed in the site. The gateway then used MQTT to disseminate these data to the cloud. The collected data can be viewed on a web-dashboard in real-time. A picture of the installation is shown in Figure 11. The power management unit continuously monitored the capacitor voltage and enableed the sensing and transmission unit when the voltage reaches the higher threshold value (3.3 V). The DASH7 transceiver module on the unit disseminated the transmitted packet.

The daily charging-discharging cycles of the supercapacitor are shown in Figure 12 and those used for a single day are depicted in Figure 13. The discharge curves in the daily cycle are due to sensing and transmission unit activities, as well as the self-discharge of the supercapacitor.

As stated already, the sensing and radio unit was only turned when the capacitor voltage reached 3.3 V. Then, a sensing task, followed by a DASH7 transmission, were executed. State 1 in Figure 13 shows an instance of sensing and transmission. Further, State 2 represents a situation where the voltage across the capacitor is below the higher threshold. Therefore, only the switching regulator is active. Along with the switching regulator’s quiescent current, the leakage current of the capacitor contributes to the discharge. States 3–8, again represent sensing, packet transmission and subsequent capacitor charging and discharging processes.

The packets transmitted by the target node were received by a DASH7 gateway and the number of packets received was recorded, as shown in Figure 14. On average, the target node was able to transmit three packets per day for the duration of the experiment. The transmissions were opportunistic and each transmission was associated with a sensing task. It must be considered that the target node was only triggered when the capacitor voltage level reached 3.3
V, and the boost converter was disabled for any voltage below 2.4
V. By bringing this threshold value to a lower level or by increasing the hysteresis, we can increase the chances of sensing and transmission. However, this might drain the capacitor more often, and hence increase the node’s downtime. We used these thresholds to ensure that at least one cycle of sensing and transmission could be performed without completely draining the capacitor. Since the buck-boost converter can work with a voltage as low as 1.8
V, the lower threshold can go down to 1.8
V. This provides a hysteresis of 1.5
V or 57.37
mJ when the upper threshold is set at 3.3 V.

Along with the packets received, we calculated the amount of energy that was harvested by the system from the capacitor charging-discharging curves. The energy harvested each day is shown in Figure 15. It must be noted that the harvested energy represents only the energy produced in daytime, as the harvester circuit does not have a polarity reversal circuit to handle the TEG output polarity change. Although there are some blackout days for the device, the harvester was still able to produce an average of 178.74 mJ of energy for the duration of the experiment, with the highest energy 696.49 mJ produced on 22 March. At present, the sensing and transmission are opportunistic. However, it might be more reliable to execute sensing frequently and transmissions only when there is plenty of energy available. Assuming the lower threshold of the comparator is set at 1.8
V, with the harvested energy, the device can execute at least 100 sensing tasks and five transmissions per day. However, the self-discharge of the supercapacitor poses a challenge. Since the capacitor energy discharges in a short time, saving the leftover energy for a low-harvesting day seems nearly impossible.

## 6. Discussion and Future Work

We have identified the areas of improvement and plan to work on this further. One of the major areas of focus would be to improve the thermal efficiency of the energy generator unit. The TEGs, in general, are highly inefficient at lower temperatures. The addition of an extra element to the system adds to the inefficiency, which, in turn, reduces the output power from the TEG. Hence, the heat conduction path provided for the hot and cold sides of the TEG must add minimal loss to the system. Thus, we are evaluating the current design of the energy harvester unit to figure out ways to improve its performance. We are also looking into the possibility of using a different structure, such as a buried heat sink, as was proposed in [20], and analysing each structure’s thermal efficiency. We plan to benchmark different structures by emulating their performance for low-temperature gradients. Along with this, we are looking at alternative TEG modules, which provide a better performance than the currently used TG12-4-01L. For instance, generator TG12-8-01LS [38] from the same manufacturer provides almost double the power output as TG12-4-01L, but at an increased form factor. Additionally, there are many TEG modules available on the market at present, but few modules’ datasheets provide details on their behaviour at extremely low-temperature gradients below 10 °C. Therefore, we plan to characterize TEG modules for temperature gradients below 10 °C with the help of an emulation tool.

The harvester circuit can currently only harvest energy during the daytime, as the output of the TEG modules changes polarity at night due to changes in air temperature. Therefore, there is a requirement for an auto-polarity detection circuit, which can reverse the input polarity without compromising the efficiency of the system. We are considering the addition of a low-power rectifier circuit similar to [21], or a different harvester IC such as LTC3109, which comes with in-built polarity detection.

The TEG module is essentially a Seebeck element, which can be used to measure the temperature of one side if the temperature of the other side is known. This implies that the open-circuit voltage of the TEG can be used to calculate differences in temperature and the temperature on either sides, if the other side is known. Hence, we are considering the possibilities of exploiting the energy-converter unit for soil-temperature sensing. This would eliminate the need for a dedicated soil temperature sensor. However, this pauses certain challenges and limitations. Firstly, both the cold and hot sides of the TEG are not directly connected to the soil and air, but are connected through heat-conducting mediums such as heatsink and copper rods. As a result, the cold-side and the cold-side temperatures and, consequently, the temperature difference between the faces of the TEG may not reflect the actual temperature of soil and air. Thus, we will have to derive the heat-conduction losses through the copper rod and heat sink and compensate for these in the calculations. Secondly, we should know the temperature of the air-side in advance to calculate the soil-side temperature. Although we have an ambient temperature sensor in the device, the measurements of this sensor are not necessarily the same as the temperature of the TEG face. Thus, this would require the establishment of a relationship between the ambient-temperature sensor values and the actual temperature of the TEG air-side. In essence, the whole setup would require a pre-calibration or even on-site calibration to ensure measurement accuracy.

A major concern with any supercapacitor-based storage system is its leakage current. During the experiments, we observed a loss of almost 80 mJ of energy over a 12 h period due to the self-discharge of the super caps. This makes it difficult to shelve the leftover energy for later use. Although the backup time of the storage can be increased with a higher value capacitance, the charging time will proportionally increase. Thus, there is an increased downtime for the load, when the capacitor is fully discharged. To solve these issues, we are looking into other supercapacitors with reduced leakage current, as well as alternative storage systems.

The current design employs an opportunistic measurement and transmission logic. Whenever enough energy is available, the sensor node is triggered. The node then takes and transmits a measurement. This increases the chances of failure and reduces the performance by frequently discharging the capacitor, especially when the temperature difference is very low. However, in most of these environmental sensing applications, the real-time reporting of data is not really important. The data can tolerate delay, and hence could be buffered. This allows for us to reduce the number of transmissions, since we can transmit packets when there is enough energy for the process. This cannot only reduce the stress on the capacitor, but also reduces the overall energy consumption of the sensor node by reducing the number of on and off state changes in the radio module.

We are also considering the applicability of capacitor banks, which can be used as a variable capacitor to store the energy. This can be helpful in situations when the energy generator produces very little energy. Since the sensor nodes can only be turned on after the capacitor reaches a certain threshold value, the operations of the nodes would be blocked until the capacitor is charged to the threshold. Therefore, the charging time of the capacitor increases with a decrease in the energy converter output power. Thus, a constant capacitance will increase the downtime of the load when the charging current is low. Instead, if we have a dynamically adjustable capacitance, the system can switch to the most optimal storage value so that the downtime is minimal. This would further require an intelligent algorithm to be implemented on the power management side so that the incoming energy can be tracked and predicted. Based on this, the system can dynamically adjust the threshold values and capacitance.

## 7. Conclusions

We have demonstrated the possibility of harvesting energy from soil–air temperature differences to power a DASH7 environment sensor. The proof-of-concept design and positive real-life evaluation prove that the soil–air temperature difference could be an alternative energy source for low-power environmental sensing systems, especially in situations where continuous solar radiation is not available or the reach of the radiation is limited due to vegetation cover or other obstacles. During the duration of the experiment, the device was able to harvest, on average, 178.74
mJ of energy, with the highest amount of energy harvested being 696.49 mJ. The energy harvested by the harvester circuit is enough to intermittently charge a 15 mF supercapacitor. Furthermore, this energy is enough to perform at least 100 sensing tasks and five transmissions each day. Additionally, since we are using DASH7, which has a similar energy footprint to LoRaWAN, we expect the system to also work seamlessly with LoRaWAN. However, a blackout period is observed every day when the air and soil temperature presents a concern. We plan to work on this further and devise a solution to tackle this problem. We further plan to take the harvester design to the production level and to install more energy-harvesting devices in our research sites in Belgium and Iceland.

## Figures and Tables

**Figure 1 sensors-22-04737-f001:**
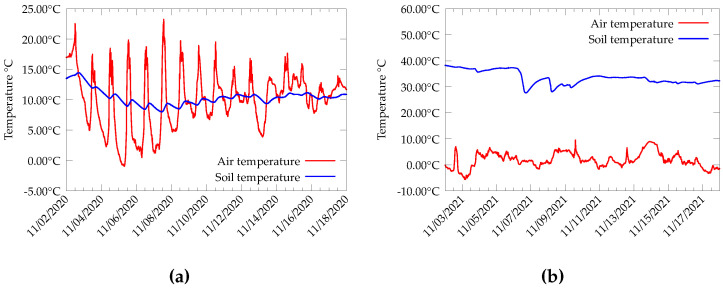
Soil and air temperature data collected from (**a**) Campus Drie Eiken of University of Antwerp in Belgium for the month of November 2020. (**b**) Forhot site in Iceland for the month of November 2021.

**Figure 2 sensors-22-04737-f002:**
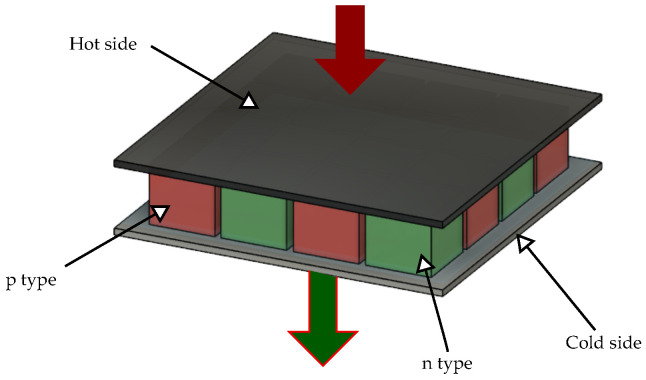
3D view of a Thermoelectric Generator.

**Figure 3 sensors-22-04737-f003:**
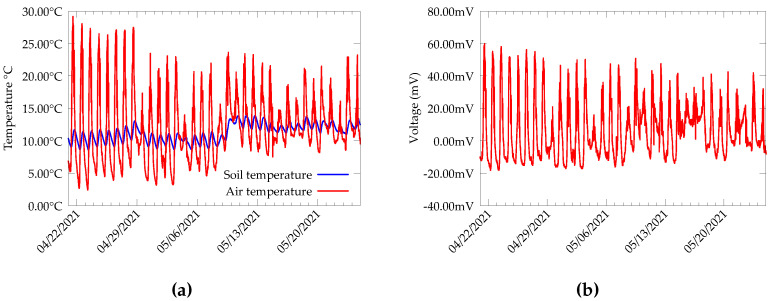
Data from test bed deployed in the CDE campus of University of Antwerp during April 2021. (**a**) Soil temperature (TEG cold side) and air temperature (TEG hot side). (**b**) TEG open circuit voltage.

**Figure 4 sensors-22-04737-f004:**
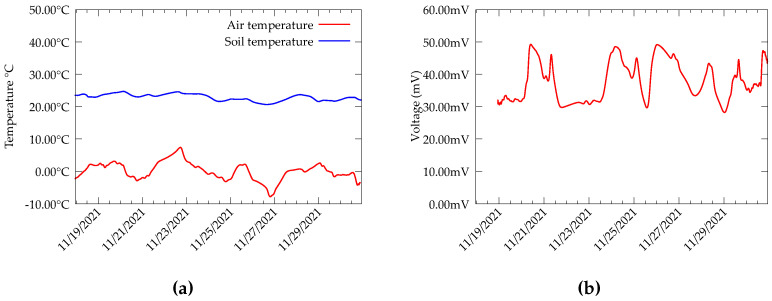
Data from test bed deployed in Forhot reseach site, Iceland. (**a**) Soil temperature (TEG hot side) and Air temperature (TEG cold). (**b**) TEG open circuit voltage.

**Figure 5 sensors-22-04737-f005:**
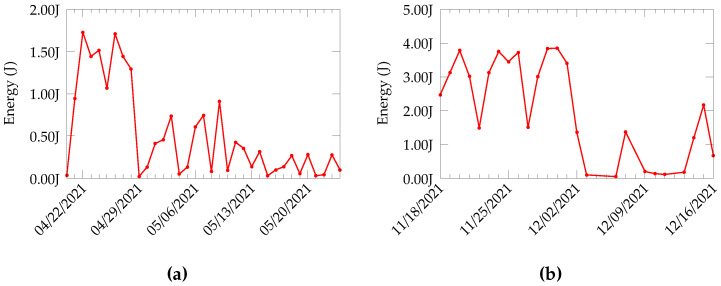
Energy estimation based on data collected from CDE and Forhot sites (**a**) Estimated values for CDE. (**b**) Estimated values for Forhot.

**Figure 6 sensors-22-04737-f006:**
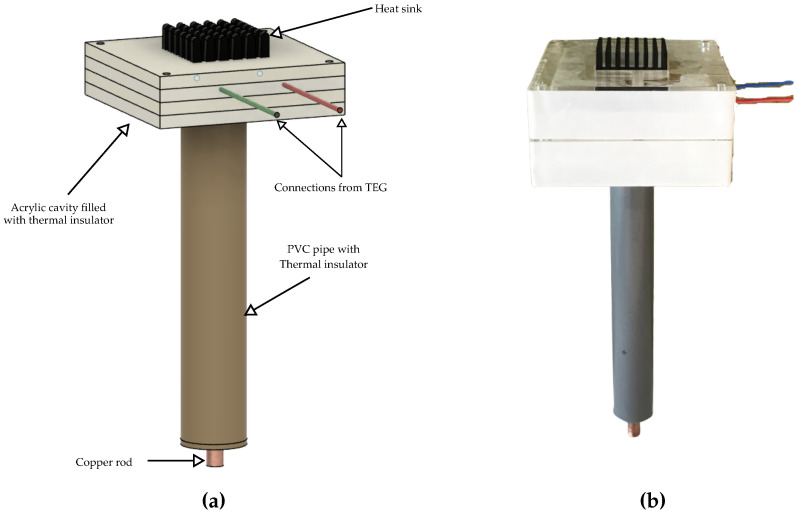
Design of energy converter unit (**a**) 3D model (**b**) Fabricated unit.

**Figure 7 sensors-22-04737-f007:**
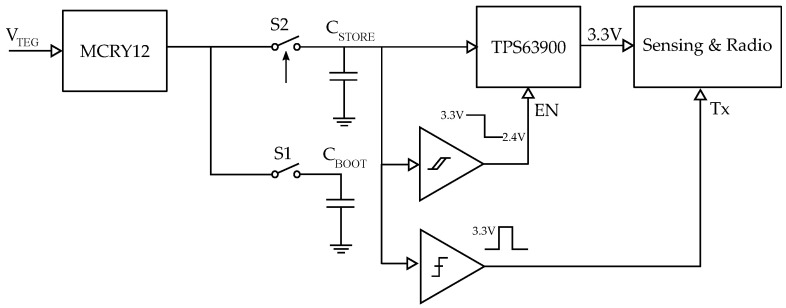
Block diagram of the energy harvester power management unit.

**Figure 8 sensors-22-04737-f008:**
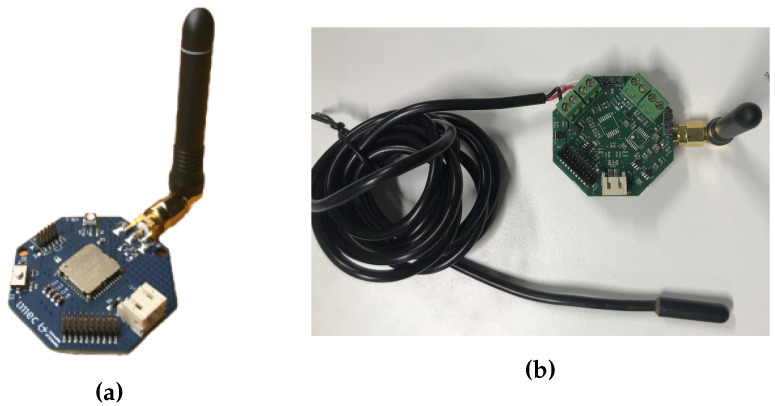
Sensing and radio unit (**a**) Dash7 radio module. (**b**) Sensing unit connected to Dash7 radio module.

**Figure 9 sensors-22-04737-f009:**
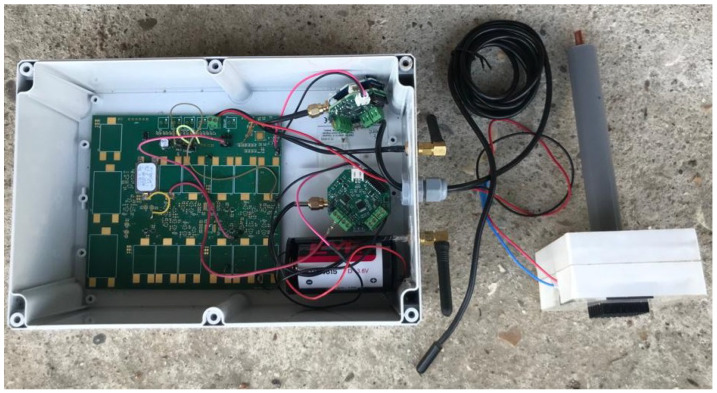
A finished unit with data logger assembled in IP66 ABS enclosure ready for deployment.

**Figure 10 sensors-22-04737-f010:**
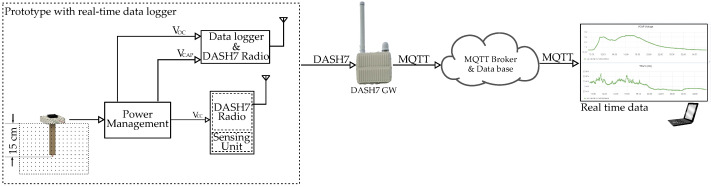
A block diagram of the complete experiment setup with harvester, data-logger and real-time data on a dashboard.

**Figure 11 sensors-22-04737-f011:**
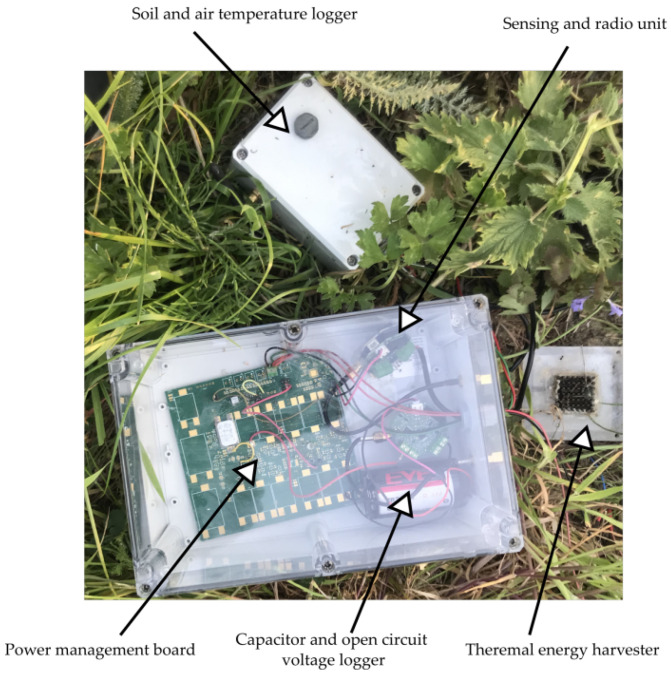
Energy-harvester system deployed in the CDE Campus of University of Antwerp in Belgium.

**Figure 12 sensors-22-04737-f012:**
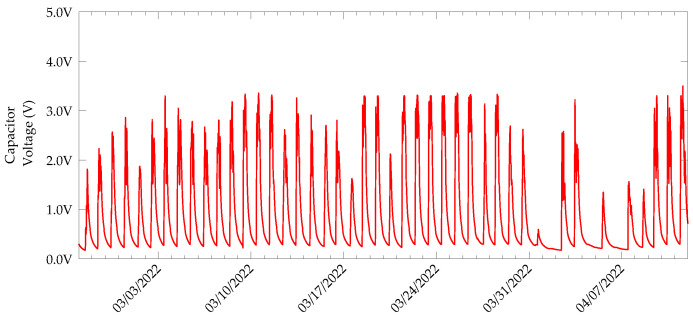
Voltage across C_STORE_ recorded in every one minute for the whole duration of the experiment.

**Figure 13 sensors-22-04737-f013:**
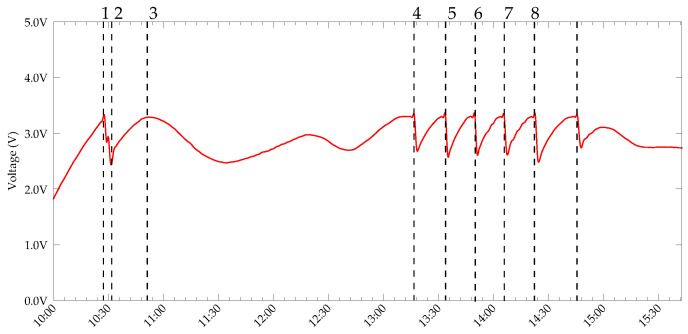
Capacitor voltage recorded for a single day.

**Figure 14 sensors-22-04737-f014:**
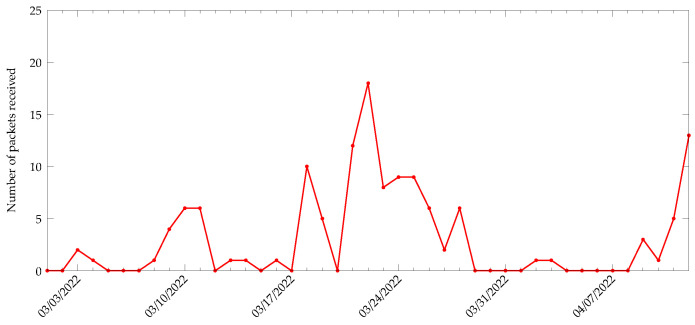
Number of packets received per day from the energy-harvesting node during the period from 1 March 2022 to 12 April 2022.

**Figure 15 sensors-22-04737-f015:**
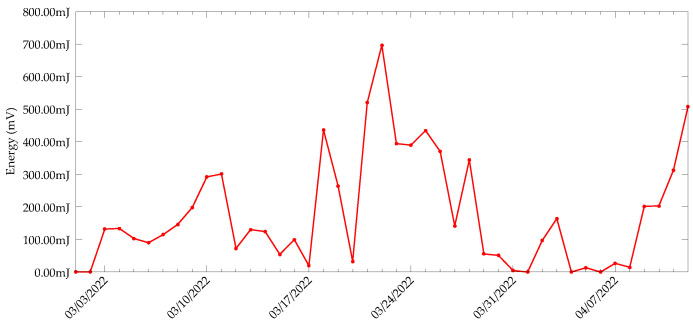
Energy harvested from the harvesting system installed in the CDE campus of University of Antwerp.

## Data Availability

The dataset presented and used in this study are openly available in Zenodo at DOI:10.5281/zenodo.6687399; Link https://zenodo.org/record/6687399#.YrOS6nZBw2w (accessed on 24 April 2022).

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
