# Peer review of "Battery-Less Environment Sensor Using Thermoelectric Energy Harvesting from Soil-Ambient Air Temperature Differences"

_sensors, 2022, doi:10.3390/s22134737_

Round 1

Reviewer 1 Report

The work is well structured and presented. After presenting the harvesting technology, the authors evaluate the harvestable energy potential and design the autonomous sensor. The choice of components is justified each time.
The authors should provide some additional information in the state of the art section on the level of consumption of the DASH7 protocol. It is mentioned in line 168 that the level of consumption of the DASH7 protocol is comparable to that of the LoRAWAN protocol; it would be interesting to provide other consumption levels apart from that of the reference in [24].

Reviewer 2 Report

General remarks:
This is a very well given paper, some minor comments:
1.    The abstract should overview the main results and claims. 
2.    How the paper differs from other publications? Is the performed system, or which part is a new contribution to the paper?
3.    There is a lack of the functional scheme of the energy converter unit.
4.    Also, the scheme of the full system consisting of sensor/converter, wireless node and power management should be presented.
5.    What are the ways to increase the efficiency of the generator to harvest more energy?
6.    Is possible to use this energy generator as the temperature sensor itself?
7.    There is no analytical/theoretical input in the paper according to the system design.
8.    The conclusions should be supported with results.

I invite the Authors to revise your paper.

Reviewer 3 Report

1. Both thermoelectric generators and supercapacitors have long been applicated in WSNs. The author should summarize the advantages of their own designs compared to previous similar literature.

Ref:

Verma G, Sharma V. A novel thermoelectric energy harvester for wireless sensor network application[J]. IEEE Transactions on Industrial Electronics, 2018, 66(5): 3530-3538.

Lin Q, Chen Y C, Chen F, et al. Design and experiments of a thermoelectric-powered wireless sensor network platform for smart building envelope[J]. Applied Energy, 2022, 305: 117791.

2. The wireless communication module is a mature and commercial one, without any special design tricks and new contributions.

3. The power management unit is very simple.

4. The performance for energy density should be compared quantitatively with the literature to determine if there are any advantages.

5. Does the energy harvesting performance depends on how deep the copper rod penetrates into the soil?

Overall, this manuscript lacks sufficient design tricks and performance comparisons.

Round 2

Reviewer 2 Report

The block diagram scheme of the whole system should be provided, as I have pointed out in the first review.

Reviewer 3 Report

The authors failed to convince me on any issues that I concerned in my last comment. The work in the manuscript is more like the design of industrial products than scientific research. In fact, this work is mainly a combination of many mature technologies. I believe that an industrial product design can also be a good paper if and only if it proposes and solves a well-formulated optimization problem with practical values. However, this manuscript fails in this aspect. To sum up, I think this manuscript is unsuitable to be published on sensors.

Author Response

Dear Reviewer,

It is unfortunate that we couldn't convince you of the contributions that our work makes to the research of thermal energy harvesting. We appreciate your time and feedback. However, we would like to make a couple of points- we are definitely solving a problem; we mention that clearly in the introduction. The whole point of utilizing the soil thermal energy for powering our sensors is to ensure that they keep operating even in environments where solar harvesters fail. Especially in applications like environmental sensing where they are prone to performance decay over time due to dust and vegetation cover. Secondly, for the first time, we integrate DASH7 communication with the soil energy harvesters. Further, we are sure that if DASH7 works, then LoRaWAN will also work. Nonetheless, we are not using LoRaWAN for now as our existing deployment does not use LoRaWAN.  We are definitely not claiming it to be fine-tuned industrial product. But, it is indeed a small step toward a more sophisticated system.

Thank you for your understanding!